# Optimization of Blended Biochar Pellet by the Use of Nutrient Releasing Model

**JoungDu Shin [1],\* and SangWon Park [2]**

1    Department of Climate Change and Agro-Ecology, National Institute of Agricultural Sciences,
     WanJu Gun 55365, Korea
2    Chemical Safety Devision, National Institute of Agricultural Sciences, WanJu Gun 55365, Korea;
     swpark@korea.kr
\*    Correspondence: jdshin1@korea.kr; Tel.: +82-63-238-2494; Fax: +82-63-238-3823

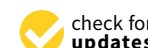

**Featured Application: A potential application is to develop a slow release fertilizer of biochar pellet type based on selected optimum biochar pellet.**

**Abstract:** For the recycling of biomass conversion materials, this experiment was conducted to investigate plant nutrient releasing characteristics, and to determine an optimum blended ratio of biochar for producing a biochar pellet based on a column leaching study. The treatments consisted of only pig manure compost (PMC) as a control, pig manure compost pellets (PMCP), and biochar pellets (BCP) blended with biochar and pig manure compost with the following ratios: 9:1, 8:2, 4:6, and 2:8. Results showed that the accumulated amount of ammonium nitrogen ($NH_4$-N) was in order of PMC > PMCP > BCP (2:8) > BCP (4:6) > BCP (8:2) > BCP (9:1) ratios. The highest accumulated amounts of phosphate phosphorus ($PO_4$-P) and potassium (K) were 1953 and 1917 mg $L^{-1}$ in the PMC and PMCP, but the lowest in the BCP (9:1) were 223 and 1078 mg $L^{-1}$, respectively. It was shown that the highest accumulated amount of silicon dioxide ($SiO_2$) was 2329 mg $L^{-1}$ in the BCP (8:2), but the lowest in the PMC was 985 mg $L^{-1}$. The estimations for accumulated $NH_4$-N, $PO_4$-P, K, and $SiO_2$ releasing amounts in all the treatments were significantly fitted with a modified Hyperbola model. The optimum mixing rate was estimated to be BCP (2:8). Therefore, biochar pellets might be useful in obtaining basic information on slow-release fertilizer for sustainable agriculture.

**Keywords:** biochar pellet; modified Hyperbola model; nutrient release; pelletization; pig manure compost

## 1. Introduction

Technology of carbon sequestration (C seq.) and mitigation of carbon dioxide equivalency ($CO_2$-*equiv.*) emissions need to be developed by using by-products of agricultural biomass through carbon-recycle systems in cropland. By-products from agricultural biomass consist of carbonaceous materials such as rice hulls, crop residues, trimming branches, animal waste, and bio-waste from fruit and vegetable markets. In Korea's agricultural sector, the total potential biomass production is estimated at 58,010 Gg $yr^{-1}$ [1], which can be converted into a less non-degradable form through biomass conversion technology.

Biochar, a porous and carbonaceous material obtained from biomass conversion with thermal treatment under limited oxygen, is one practical option for soil carbon sequestration. It contains a non-degradable structural carbon with double bonds and an aromatic ring that cannot be broken down by microbial organism activities [2]. The produced biochar could be utilized for several purposes [3] However, in its application, 30% is lost due to wind, while 25% is lost during spreading in cropland [4]. On the other hand, one disadvantage of biochar application in the field

is the presence of fine dust caused by wind during the spreading of biochar, which could affect farmer's respiratory organs.

The application of 5% biochar produced at 700 °C had the most significant C seq. during rice and leaf beet cultivation [5]. For cropland C seq., Shin et al. [6] reported that C seq. is highest at 2.3 tons ha$^{-1}$ in corn fields incorporated with biochar and cow manure compost. The mitigation of $CO_2$-*equiv.* emission is estimated at 7.3 to 8.4 T ha$^{-1}$ and profits ranged from \$57 to \$163 when incorporated with 2600 kg ha$^{-1}$ of biochar in corn fields.

Biochar's effects in agro-ecology have been suggested to come from the plant's sorption and retention abilities of available nutrients [7]. Biochar from holm oak tree (*Quercus ilex*) increased ammonium nitrogen ($NH_4$-N) adsorption in sandy acrisol associated with humid and tropical climates, but had no effect on nitrate nitrogen ($NO_3$-N) sorption in the column experiment [8]. However, the largest amount of sorption of $NH_4$-N and binding strength, constant for biochar derived from rice husks, have been calculated as 0.5 mg L$^{-1}$ and 0.03 mg L$^{-1}$, respectively [9]. Shin [10] reported that rates of nitrogen (N) mineralization and nitrification are low in corn fields incorporated with biochar, compared to plots with different organic composts alone. These could be attributed to the sorption capacity of $NH_4$-N to biochar.

Phosphorous (P) and potassium (K) are essential elements for both crop growth and the maintenance of crop productivity [11,12]. The maximum sorption amount and binding strength constant of $PO_4$-P are estimated at 0.1 mg L$^{-1}$ and 0.06 mg L$^{-1}$, respectively, for biochar derived from oak tree [13]. K deficiency in soil is mainly due to presence of 90% to 98% of insoluble K forms or due to the high soluble form of the available K [14–17]. Crop residues of rice and sugarcane contain applicable amounts of silicon (Si) [18,19]. Si plays an important role in plant cell wall strength and insect defense, as well as nutrient uptake improvement [20–22]. Plant silicon is thought to be a recycling Si pool that can be accumulated in surface soil after litter fall and recovered from plant decomposition [23].

A biochar pellet is one option to reduce fine dust and biochar loss by strong wind and intensive rainfall, thus decreasing handling and storage costs [24]. For soil incorporation, poultry litter was mixed, pelletized, and slowly pyrolyzed to produce biochar pellets [25]. Shin et al. [26] indicated that biochar pellets mixed with organic compost could be a promising option for soil C seq. and control of major plant nutrients during crop cultivation. Biochar pellets mixed with various ratios of pig manure compost was induced, and its sorption capacity and kinetic models have already been investigated [26]. For sorption test of $NH_4$-N with various loading rates, it has been shown that the maximum sorption of $NH_4$-N in the biochar pellet is 2.94 mg g$^{-1}$ where lettuce yield increased at approximately 13% relative to the control. However, there is little information on major plant releasing nutrient from the biochar pellet during leaching periods.

For the releasing model, Loney and Tabatabaie presented the leaching behavior of heavy metals from solidified and stabilized forms of biofilms using Michaelis–Menten kinetics [27]. The used model predictions confirmed that Michaelis–Meten-type kinetics is probably the most dominant mechanism for the leaching of heavy metals from cement based waste forms. Furthermore, Michaelis–Menten kinetics has been used to explore the nitrogen deposition and climate change with laboratory manipulations [28].

It is hypothesized that (1) blended biochar pellets could prolong the nutrient releasing period, and could be affected by their nutrient releasing characteristics. In addition, (2) tests should be conducted to see whether biochar pellets fit a modified Hyperbola model.

Therefore, this experiment was conducted to investigate the plant nutrient releasing characteristics, and to determine an optimum blended rate of biochar for the production of biochar pellets using a modified Hyperbola model.

## 2. Materials and Methods

### 2.1. Biochar Pellet Production

Biochar from rice hull was collected from a local farming cooperative society in Go-Chang, JenBok, Korea. The pig manure compost was purchased from the company (NOUSBO Co., Suwon, Korea) having a nationwide distribution network. The biochar was produced from a pyrolysis system and applied with "top to bottom method". This pyrolysis system consisted of rice hull burning from the upper part, thus almost excluded oxygen from outside of the system. The loading volume of rice hull was 1.5 $m^3$, and the temperature of pyrolysis process ranged from 400 to 500 °C during 4 h. The produced biochar was ground in a grinder and roller to pass through a 2 mm sieve before analysis. Physiochemical properties of biochar and pig manure compost used are presented in Table 1. The biochar was generally alkaline in nature (pH 9.8) and low in total nitrogen (TN) (2.0 g $kg^{-1}$). The content of total hydrogen (T-H) and H: C ratios were 17.6 g $kg^{-1}$ and 0.031, respectively.

**Table 1.** Characteristics of biochar and pig manure compost used [1].

| Parameters | Units | Biochar | Pig Manure Compost |
|:---:|:---:|:---:|:---:|
| pH | - | 9.78 (1:20 ratio) | 8.77 (1:5 ratio) |
| EC | dS $m^{-1}$ | 16.5 | 3.4 |
| TC | g $kg^{-1}$ | 566.0 | 289.0 |
| TOC | g $kg^{-1}$ | 553.5 | 259.0 |
| TIC | g $kg^{-1}$ | 42.5 | 30.2 |
| N | g $kg^{-1}$ | 2.0 | 29.1.0 |
| P | mg $kg^{-1}$ | 132.9 | 1744.7 |
| K | g $kg^{-1}$ | 20.2 | 14.5 |
| Si | mg $kg^{-1}$ | 1096.0 | 403.6 |

[1] TC; Total carbon, TOC; Total organic carbon, TIC; Total inorganic carbon, N; Total nitrogen, P; Total phosphorus, K; Total potassium and Si; Total silica.

Prior to pelletizing, biochar was sieved using a series of sieves (0.5 mm to 5 mm) to measure the particle distribution. It was then blended with pig manure compost as a binder to produce a biochar pellet. The combination ratios of biochar and pig manure compost were 9:1, 8:2, 6:4, and 2:8 (w/w), and the size of biochar pellet was Ø 0.51 cm × 0.78 cm. The blended materials (total weight = 2.5 kg) were thoroughly mixed using an agitator (SungChang Co., KyungGi, Korea) for 5 min. Then, while continuously mixing, the combination was sprayed with 1000 mL of deionized water for 10 min. The biochar pellet produced through the pellet machine (7.5 KW, 10 HP, KumKang Engineering Pellet Mill Co., DaeGu, Korea) with the combination of biochar and pig manure compost is described in the Figure 1.

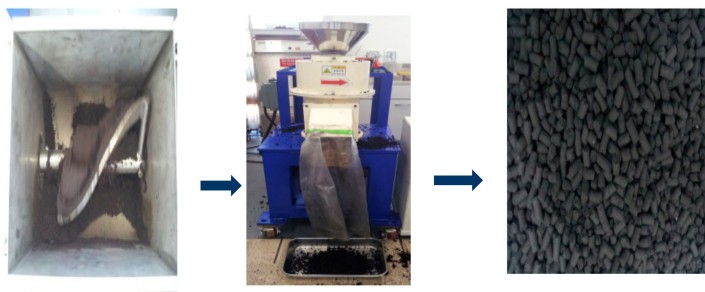

**Figure 1.** Processing diagram of the biochar pellet with different combinations of biochar and pig manure compost described by Shin et al. (2018) [26].

### 2.2. Batch Experiment for Nutrient Leaching Test

The treatments consisted of pig manure compost (PMC) as a control, pig manure compost pellet (PMCP), and various ratios of biochar pellets (BCP) blended with 2:8, 4:6, 8:2, and 9:1 of biochar/pig manure compost (w/w) in order to test the feasibility of developing a slow-release fertilizer.

For the nutrient releasing experiment, the size of the glass column was $\varnothing$ 24 mm $\times$ 40 cm with each column filled with 5 g of PMC, PMCP and different blended BCP, respectively. The column was poured into 50 mL of deionized water, completely drained, and then immediately refilled after certain retention time. The water samples were collected at 50 mL of drained water through the column until 84 days of leaching periods.

### 2.3. Chemical Analysis

The biochar and pig manure compost was taken to the National Institute of Agricultural Sciences (NIAS) to analyze the chemical properties. The pH and EC (electrical conductivity) of the biochar and pig manure compost was measured using a pH/EC meter (Orion 4 star, Thermo scientific, Singapore) at a 1:20 solid/water ratio (biochar:de-ionized $H_2O$) after shaking for 30 min in a water bath (P/NTS-3000, Eyela, Kyoto, Japan) at 140 rpm. The analytical chemical properties, such as total carbon (TC) and total organic carbon (TOC), were analyzed by a TOC analyzer (Elementar Vario EL II, Hanau, Germany) for biochar and pig manure compost. Total hydrogen was analyzed by Elemental Analyzer (Vario MACRO cube, Elementar, Langenselbold, Germany). Total P, K, and Si in the biochar and pig manure compost were measured by inductively coupled plasma atomic emission spectrometry (ICP-AES, IntegraXL, GBC LTd., Braeside, Australia) after samples were digested with nitric and hydrochloric acids.

The collected water samples were filtered using Whatman #2 filter paper, and then analyzed for $NH_4$-N, $PO_4$-P, K, and $SiO_2$ by using a UV spectrophotometer (C-Mac Co., Jenmin Dong, Dae-Jen, Korea) [26] through whole leaching periods.

### 2.4. Releasing Models

Michaelis–Menten, a general used model of substrate based kinetics, describes a saturating function of substrate concentration with parameters $V_{max}$, the maximum reaction velocity, and kM, the half saturation constant, which corresponds to the substrate concentration [S] when $V_{max}/2$. This model was used to predict the accumulated amounts of released material as a function of leaching periods. Therefore, a modified Hyperbola model from Michaelis–Menten equation used is shown below;

$$Y = Amax [t]/(t1/2_{(Amax)} + [t])$$

Y: accumulated concentration (mg $L^{-1}$); Amax: maximum accumulated concentration (mg $L^{-1}$); t1/2$_{(Amax)}$: required time to reach 1/2 Amax; t: leaching periods (days).

### 2.5. Statistical Analysis

The statistical analyses for total water-soluble amounts of $NH_4$-N, $PO_4$-P, K, and $SiO_2$ were performed using a one-way ANOVA with 6 levels, using SAS version 9.0 (SAS Institute, Carry, NC, USA). Duncan multiple range test was used for accessing significant differences ($p < 0.0001$) among treatment means during leaching periods. Means of variables were compared with parameters among treatments by using the above equation according to $p$-values $< 0.0001$ after analysis of variances (ANOVA). The validity of a modified Hyperbola model for each parameter was assured for normal distribution by Shapiro–Wilk test ($p < 0.05$). The releasing model for each nutrient was established by data analysis using SigmaPlot 12 (Systat Software, Inc., San Jose, CA, USA). The model used was calculated using the equation based on correlation coefficient values ($R^2$).

## 3. Results

It was observed that the more biochar contained within the biochar pellet, the greater sorption of $NH_4$-N. For accumulated $NH_4$-N releasing amount, the order was PMC > PMCP ≥ BCP (2:8) > BCP (4:6) > BCP (8:2) > BCP (9:1) ratios, and the highest accumulated amount in the PMC was 397 mg $L^{-1}$ (Figure 2).

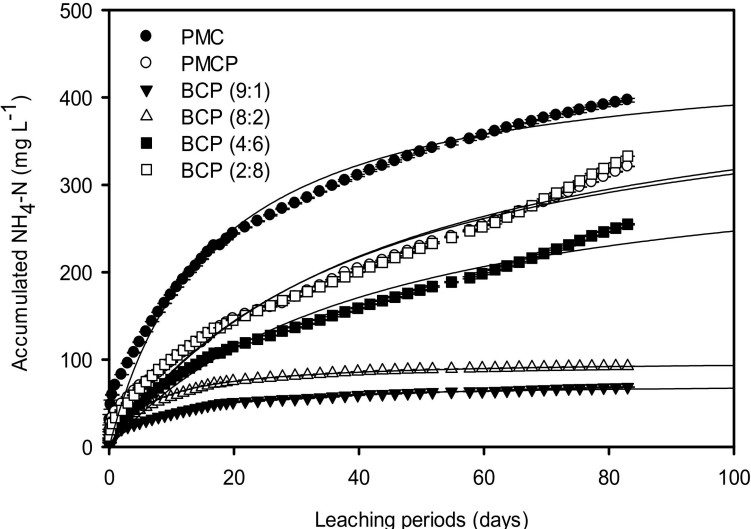

**Figure 2.** Effects of $NH_4$-N accumulated concentrations in the releasing water from biochar pellet blended with various ratios of pig manure compost during leaching periods.

The estimation parameters for an accumulated $NH_4$-N releasing amount from different types of biochar pellets are presented in Table 2. The model was significantly correlated with the $R^2$ values between the observed and estimated values. The maximum accumulated $NH_4$-N releasing amount was observed in PMC even if same amount of pig manure compost between PMC and PMCP was used in the leaching column. This might be due to the increase mass of pig manure compost. However, the required time to reach half of the maximum accumulated amount in PMC was taken only 16 at days, but was 41 days in PMCP. Therefore, pelletization could reduce the releasing rates of $NH_4$-N even if the same loading amount of material was used. The greater accumulated amount and longer leaching periods were observed in the BCP (2:8 and 4:6). The estimated releasing model was significantly fitted with all treatments (Table 2).

**Table 2.** Estimation model for accumulated $NH_4$-N releasing from different combination rates of biochar pellets.

| Treatments | Model Parameters | | | | Analysis of Variance | | $R^2$ |
|---|---|---|---|---|---|---|---|
| | $A_{max}$ | *p*-Values | $t_{1/2(Amax)}$ | *p*-Values | F | *p*-Values | |
| PMC | 455.7 | <0.0001 | 16.5 | 0.0003 | 2375.7 | <0.0001 | 0.98 |
| PMCP | 441.3 | <0.0001 | 41.2 | <0.0001 | 1747.7 | <0.0001 | 0.97 |
| BCP (9:1) | 72.7 | <0.0001 | 8.2 | <0.0001 | 2044.5 | <0.0001 | 0.97 |
| BCP (8:2) | 99.3 | <0.0001 | 6.5 | <0.0001 | 2341.1 | <0.0001 | 0.98 |
| BCP (4:6) | 350.9 | <0.0001 | 42.0 | <0.0001 | 2247.0 | <0.0001 | 0.98 |
| BCP (2:8) | 455.5 | <0.0001 | 43.5 | <0.0001 | 1607.4 | <0.0001 | 0.97 |

PMC; control as pig manure compost, PMCP; pig manure compost pellet, and BCP; ratios of biochar pellets blended of biochar/pig manure compost 2:8, 4:6, 8:2, and 9:1.

The accumulated amount of $PO_4$-P was not significantly different with biochar pellets which contained 0 to 40% of biochar during leaching periods. It was shown that the highest accumulated amount of $PO_4$-P in the PMC was 1953 mg $L^{-1}$ and the lowest in the BCP (9:1 ratio) was 223 mg $L^{-1}$ (Figure 3). This could be attributed to the high concentration of $PO_4$-P in the PMC. However, this could

also result in the slower release of $PO_4$-P in the PMCP than that found in the PMC because of the change in the physical characteristics in pelletization of pig manure compost.

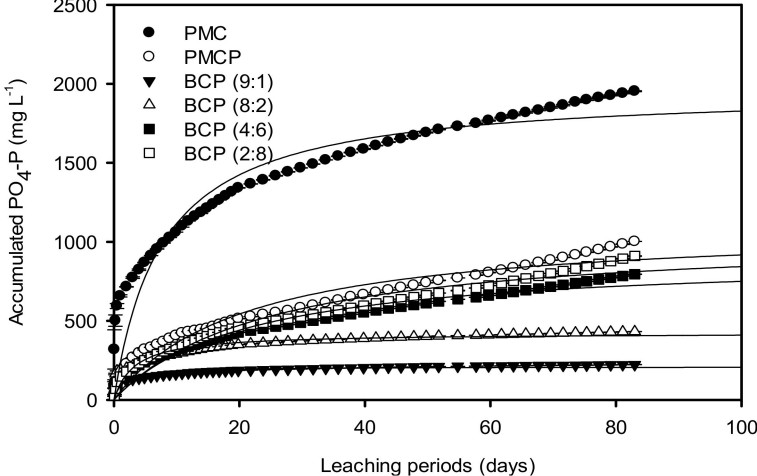

**Figure 3.** Effects of accumulated $PO_4$-P concentrations in the releasing water from biochar pellet blended with various ratio of pig manure compost.

The curves fit was derived from the estimation model calculated by a modified equation based on correlation coefficient values ($R^2$). The estimation models for accumulated $PO_4$-P releasing amount from different ratios of biochar pellets showed that estimation values from a modified equation were significantly correlated with the observed values for accumulated $PO_4$-P releasing amount in all treatments (Table 3). The required time of 1/2 maximum accumulated $PO_4$-P amount was taken 21 days in the PMCP and BCP (4:6). Similar patterns for the estimation model were observed in the accumulated $NH_4$-N releasing models (Tables 2 and 3). The estimated releasing model has significantly fit with all the treatments (Table 3).

**Table 3.** Estimation model for accumulated $PO_4$-P releasing amount from different types of biochar pellets.

| Treatments | Model Parameters | | | | Analysis of Variance | | $R^2$ |
|---|---|---|---|---|---|---|---|
| | $A_{max}$ | *p*-Values | $t_{1/2(A_{max})}$ | *p*-Values | F | *p*-Values | |
| PMC | 1966.4 | <0.0001 | 7.6 | 0.0003 | 378.8 | <0.0001 | 0.88 |
| PMCP | 1108.3 | <0.0001 | 20.9 | <0.0001 | 686.4 | <0.0001 | 0.93 |
| BCP (9:1) | 208.8 | <0.0001 | 1.4 | <0.0001 | 122.3 | <0.0001 | 0.70 |
| BCP (8:2) | 422.3 | <0.0001 | 3.3 | <0.0001 | 200.2 | <0.0001 | 0.79 |
| BCP (4:6) | 908.5 | <0.0001 | 21.2 | <0.0001 | 1092.1 | <0.0001 | 0.95 |
| BCP (2:8) | 1050.3 | <0.0001 | 24.5 | <0.0001 | 1002.5 | <0.0001 | 0.95 |

PMC; control as pig manure compost, PMCP; pig manure compost pellet, and BCP; ratios of biochar pellets blended of biochar/pig manure compost 2:8, 4:6, 8:2, and 9:1.

The highest accumulated amount of K in the PMCP was 1917 mg $L^{-1}$ and the lowest in the BCP (9:1) was 1078 mg $L^{-1}$. It appeared that accumulated amounts of K abruptly increased at the early leaching stage, and gradually increased at later stages (Figure 4).

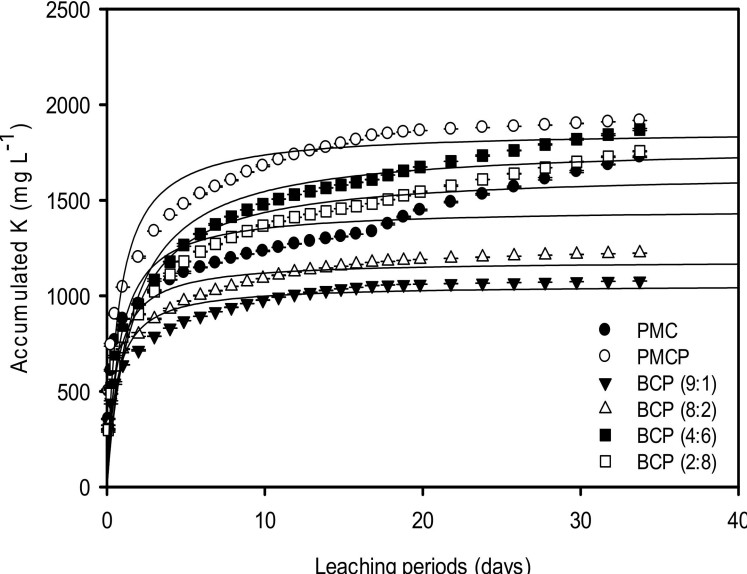

**Figure 4.** Effects of accumulated K concentrations in the releasing water from biochar pellet blended with various ratios of pig manure compost.

The estimation model for K accumulated amount from different types of biochar pellets is presented in Table 4. The estimation values calculated from the modified equation were significantly correlated with the observed values for K releasing, regardless of combinations of biochar pellets. Most of K was released within 2 days of leaching periods. The estimated releasing model for the modified equation significantly fits with all the treatments (Table 4).

**Table 4.** Estimation model for accumulated K releasing amount from different types of biochar pellets.

| Treatments | Model Parameters | | | | Analysis of Variance | | $R^2$ |
|---|---|---|---|---|---|---|---|
| | $A_{max}$ | *p*-Values | $t_{1/2(Amax)}$ | *p*-Values | F | *p*-Values | |
| PMC | 1456.1 | <0.0001 | 0.8 | 0.0003 | 94.2 | <0.0001 | 0.76 |
| PMCP | 1864.5 | <0.0001 | 0.7 | <0.0001 | 257.5 | <0.0001 | 0.90 |
| BCP (9:1) | 1056.8 | <0.0001 | 0.6 | <0.0001 | 299.5 | <0.0001 | 0.91 |
| BCP (8:2) | 1182.4 | <0.0001 | 0.6 | <0.0001 | 259.6 | <0.0001 | 0.90 |
| BCP (4:6) | 1789.9 | <0.0001 | 1.6 | <0.0001 | 346.3 | <0.0001 | 0.92 |
| BCP (2:8) | 1649.6 | <0.0001 | 1.5 | <0.0001 | 302.0 | <0.0001 | 0.91 |

PMC; control as pig manure compost, PMCP; pig manure compost pellet, and BCP; ratios of biochar pellets blended of biochar/pig manure compost 2:8, 4:6, 8:2, and 9:1.

It has appeared that the highest accumulated amount of $SiO_2$ in the PMCP was 1259 mg $L^{-1}$, but the lowest in the PMC was 634 mg $L^{-1}$ at 20 days of leaching periods (Figure 5). Following this, it then decreased after that period even if it did not contain biochar from rice hull. The estimation model for accumulated $SiO_2$ releasing amount from different BCP is presented in Table 5. The estimation values calculated from a modified equation were significantly correlated with the observed values for accumulated $SiO_2$ releasing amount regardless of combination rates of biochar pellets. The estimated releasing model equation was had a highly significant fit with whole treatments (Table 5).

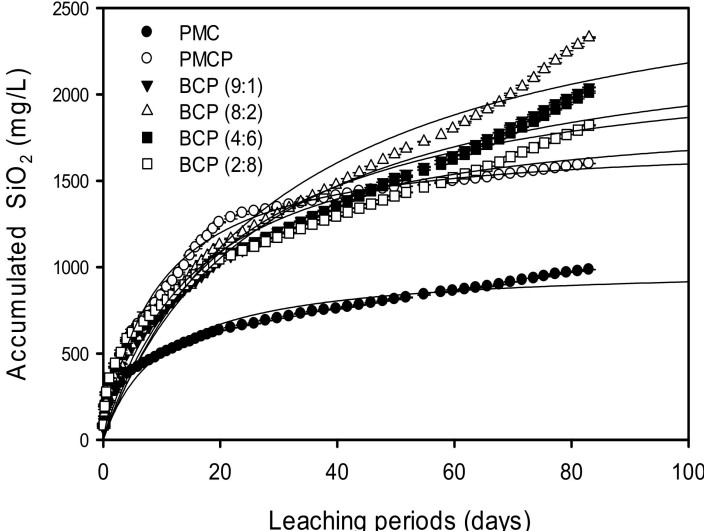

**Figure 5.** Effects of accumulated $SiO_2$ concentrations in the releasing water from biochar pellet blended with various ratios of pig manure compost during leaching periods.

**Table 5.** Estimation model for accumulated $SiO_2$ releasing amount from different types of biochar pellets.

| Treatments | Model Parameters | | | | Analysis of Variance | | $R^2$ |
|---|---|---|---|---|---|---|---|
| | $A_{max}$ | *p*-Values | $t_{1/2(Amax)}$ | *p*-Values | F | *p*-Values | |
| PMC | 1006.8 | <0.0001 | 10.1 | 0.0003 | 871.4 | <0.0001 | 0.94 |
| PMCP | 1742.1 | <0.0001 | 9.1 | <0.0001 | 3047.1 | <0.0001 | 0.98 |
| BCP (9:1) | 2445.0 | <0.0001 | 26.5 | <0.0001 | 1440.5 | <0.0001 | 0.96 |
| BCP (8:2) | 2871.6 | <0.0001 | 31.6 | <0.0001 | 1449.7 | <0.0001 | 0.96 |
| BCP (4:6) | 2277.7 | <0.0001 | 22.0 | <0.0001 | 957.3 | <0.0001 | 0.95 |
| BCP (2:8) | 1935.4 | <0.0001 | 15.5 | <0.0001 | 886.0 | <0.0001 | 0.94 |

PMC; control as pig manure compost, PMCP; pig manure compost pellet, and BCP; ratios of biochar pellets blended of biochar/pig manure compost 2:8, 4:6, 8:2, and 9:1.

It was observed that the estimation model for accumulated $NH_4$-N, $PO_4$-P, and $SiO_2$ releasing amounts in the biochar pellet was a significantly fit with this modified equation regardless of combination ratios of biochar and pig manure compost. In aspect of 1/2 releasing amount, the BCP (8:2) could be better for crop uptake and reduced application amount of Si fertilizer.

The total water soluble amounts of $NH_4$-N, $PO_4$-P in PMC were significantly higher than those of PMCP, but soluble amounts of K and $SiO_2$ in PMC were significantly lower than those of PMCP during leaching period (Table 6). However, it is observed that accumulated amounts of water soluble $NH_4$-N, $PO_4$-P, K and $SiO_2$ were usually greater in the BCP (2:8 and 4:6) compared to the PMC (Table 6).

**Table 6.** Comparisons of total water-soluble accumulated amounts of $NH_4$-N, $PO_4$-P, K, and $SiO_2$ for different treatments during leaching periods.

| Treatments | Total Water Soluble Amounts (mg) | | | |
|---|---|---|---|---|
| | $NH_4$-N | $PO_4$-P | K | $SiO_2$ |
| PMC | 397 a | 1953 a | 1727 b | 985 d |
| PMCP | 321 b | 1002 a | 1917 a | 1600 c |
| BCP (9:1) | 69 d | 223 e | 1078 b | 2041 b |
| BCP (8:2) | 92 d | 432 d | 1223 b | 2329 a |
| BCP (4:6) | 255 c | 795 c | 1870 b | 2013 b |
| BCP (2:8) | 333 b | 910 c | 1756 b | 1821 bc |
| F | 77.0 | 259.2 | 5826.9 | 46.9 |
| *p*-values | <0.0001 | <0.0001 | <0.0001 | <0.0001 |

PMC; control as pig manure compost, PMCP; pig manure compost pellet, and BCP; ratios of biochar pellets blended of biochar/pig manure compost 2:8, 4:6, 8:2, and 9:1. Means values followed by different letters indicate significant differences ($p < 0.0001$) among treatments with $NH_4$-N, $PO_4$-P, K, and $SiO_2$ (ANOVA and subsequent Duncan multiple range test).

## 4. Discussion

*4.1. Patterns of Accumulated $NH_4$-N Releasing Amount with Different Pellets and Application of a Modified Hyperbola Model*

Nitrogen fertilizers generally occur in chemical transformations such as ammonification, nitrification, denitrification, nitrogen fixation, and immobilization in soil [29,30]. Urea is widely applied as nitrogen fertilizer due to the rapid release of N to soil, from which plants use only 40%, and 60% is lost in different ways [31]. The maximum evaporative loss was estimated to be from 26.5% to 29.4%, which contributes to greenhouse gases. Therefore, a slow N releasing fertilizer is the best way to minimize $N_2O$ emission from soil [32]. In the $NH_4$-N accumulation amount during leaching periods, it was observed that the greater biochar, the less releasing amount there is. This reason for this might be attributed to the $NH_4$-N sorption in the biochar. This finding might be attributed to the previous experimental result, showing that the higher the amount of biochar contained in the pellet, the greater the sorption of $NH_4$-N [26]. It might be assumed that Mg, $NH_4$-N, and $PO_4$-P in the biochar and pig manure compost could be synthesized to struvite [Mg $(NH_4)_2PO_4 \cdot 6H_2O$] as slow-release fertilizer during processing time of biochar pellet. Prakongkep et al. (2015) reported that the soluble compounds of total Mg and total P contained 1% to 21% and 1% to 33%, respectively, in biochar [33].

For applying the releasing model of a modified Hyperbola model, it was observed that there was a basic law that greatly reduced the $NH_4$-N release from the BCP treatments compared to the PMC because of changes of its physical properties with pelletization. In addition to the biochar pellets with various combination rates of pig manure compost, the accumulated $NH_4$-N and $PO_4$-P releasing amounts in the BCP treatments decreased from 49% to 91% and from 61% to 91.3% more than those of the PMC, respectively. Therefore, these biochar pellets had more potential ability on greenhouse gas emission reduction, as well as eutrophication than that of PMC when applied to cropland for agricultural sustainability.

*4.2. Pattern of Accumulated $PO_4$-P, K and $SiO_2$ Releasing Amounts with Different Pellets and Appication of a Modified Hyperbola Model*

A large amount of nitrogen and phosphate fertilizer is applied to soil every year to increase soil fertility. Yearly, the present consumption of rock phosphorous as fertilizer is over one million tons [34]. Excessive phosphorous in lakes and ponds is a major cause of eutrophication, which occurs on a global scale and destroys aquatic ecosystems [35]. Accelerated eutrophication not only affects aquatic ecosystems, but also indirectly inhibits economic progress [36]. Muriate of potassium chloride (KCl) is applied as a dominant chemical fertilizer to supply available K to crops [37]. Si is usually taken up by plant as an uncharged monosilicic acid ($H_4SiO_4$), and polymerized silica gel ($SiO_2$-$nH_2O$) is known to be as high as 90% of soluble phytolith formed in plants [38]. Biochar application increased soil and plant available Si content in the shoot of crops [39,40]. Biochar pellet is important in improving the sustainable agricultural ecosystem by providing plant nutrients as well as carbon sequestration.

For total water-soluble accumulated amounts of major plant nutrients, there was high level of significant difference ($p < 0.0001$) among all treatments (Table 6). It was observed that the total water-soluble accumulated amounts in the BCP (2:8) were significantly highest for $NH_4$-N, $PO_4$-P, K, and $SiO_2$ except for the BCP (8:2) compared to the PMC. For applying a modified Hyperbola model, the slightly reduced releasing $PO_4$-P fitted with this modified equation from the PMCP treatment compared to the PMC according to pelletization of pig manure compost. Furthermore, even though the accumulated $PO_4$-P in the BCP (4:6) reduced by 74.4% compared to the PMC, there was not much difference due to decreased input of pig manure compost with low releasing and binding ability. In the result of estimating the K releasing model, the releasing K amounts slightly decreased from 4.3% to 45.3% in the BCP treatments except for the BCP (4:6) compared to the PMC. However, the releasing $SiO_2$ amounts in the BCP (4:6 and 2:8) increased by 21.8% and 30.3%, respectively, compared to the PMC. It was determined that greater releasing amount and longer releasing periods were observed in

the BCP (2:8 and 4:6) based on modified Hyperbola model. However, all the plant nutrients except BCP (9:1 and 8:2) need to extend the leaching periods for equilibrium state with the other treatments.

Overall, the releasing pattern of major plant nutrient for pelletization of pig manure compost could be proposed, as Shin's principal releasing law that $NH_4$-N and $PO_4$-P releasing accumulated amount in the PMCP are decreased relative to the PMC, but $SiO_2$ in the PMCP increased compared to the PMC, with modified Hyperbola model.

However, the nutrient release pattern of biochar pellet was usually highly influenced by soil environment. Therefore, there is a need to develop a slow-release fertilizer of biochar pellet type based on selected optimum biochar pellet. For further study, an assessment of agro environmental impacts for slow-release fertilizer of biochar pellet type during crop cultivation will be performed.

## 5. Conclusions

This experiment was conducted to investigate the nutrient releasing characteristics, and to determine an optimum ratio for processing biochar pellets based on a modified Hyperbola model in terms of potential mitigation of greenhouse gas emissions and carbon sequestration. For accumulation amount of $NH_4$-N releasing, the order was PMC > PMCP > BCP (2:8) > BCP (4:6) > BCP (8:2) > BCP (9:1) ratios, and the highest accumulated amount in the PMC treatment was 397 mg $L^{-1}$. For the accumulation amount of $NH_4$-N in the BCP during leaching periods, it was shown that the greater the amount of biochar contained the greater accumulated amount except for BCP (2:8). The highest accumulated amounts of $PO_4$-P and K in the PMC were 1953 and 1727 mg $L^{-1}$, and the lowest in BCP (9:1) were 223 and 1078 mg $L^{-1}$, respectively. The highest accumulated amount of $SiO_2$ in the BCP (8:2) was 1307 mg $L^{-1}$, but the lowest in the PMC was 704 mg $L^{-1}$ at 30 days of leaching periods.

For releasing model for pellets, the releasing patterns of major plant nutrients could be proposed as Shin's principal releasing law that $NH_4$-N and $PO_4$-P releasing accumulated amounts in the PMCP are decreased, but $SiO_2$ in the PMCP significantly increased compared to the PMC as control. The optimum blended rate was estimated to be BCP (2:8) for major releasing plant nutrients based on a modified Hyperbola model.

Therefore, the biochar pellet might be used for further research on slow-release fertilizer for sustainable agriculture.

**Author Contributions:** J.S. and S.P. compiled numerical tables and graphs and completed the writing of this paper. Finally, the writing was reviewed by J.S.

**Funding:** This research was funded by the National Institute of Agricultural Sciences, Rural Development Administration beyond Research Program of Agricultural Science & Technology Development (Project No. PJ013814012018).

**Conflicts of Interest:** The author declares no conflict of interest.

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
