# Peer review of "Optimization of Blended Biochar Pellet by the Use of Nutrient Releasing Model"

_applsci, doi:10.3390/app8112274_

Reviewer 1 Report

Dear authors: Attached are my comments. Please, incorporate them

Author Response

Thank you for your valuable comments.

I revised the manuscript with your detail indications.

All best,

JoungDu Shin

Reviewer 2 Report

The manuscript reported on "Optimization of Blended Biochar Pellet by the Use of Nutrient Releasing Model". The entire manuscript requires re-writing by an English writer. In its current form, it is difficult to understand as some of the sentences are not making complete sense. While the results looks solid, the poor language usage has made it difficult to read. 

Provide more information on the justification and merit of this research.

Why was the hyperbola model chosen and what actual modification was done?

The abstract is poorly written and need to be re-written.

In the introduction write C in full.

Author Response

Thank you so much for your comments.

I modified my manuscript based on your indications.

All best,

JoungDu Shin

Reviewer 3 Report

Reviewer recommends this article for publication upon satisfactorily address concerns below:

Can authors provide a rationale for choosing pig manure for pelletization?

Can authors provide brief review on prior pelletization binders reported in literature, and provide a rationale why they want to use manure based binder for pelletization?

Reviewer  recommends following experiments/data to be included in the manuscript:

Elemental composition of biochar

Elemental composition of pig manure

Particle size of biochar

Author Response

Dear Dr. Reviewer 3

Thank you so much for your valuable commnets.

I don't have enemental analysis data of pig manure compost at this time.

I modified my manuscript as you indicated except above matter.

All best,

JoungDu Shin

Reviewer 4 Report

The topic is welcome and the paper could be interesting, but at some points it is hard to read: grammar and style should be carefully revised throughout the manuscript, to improve clarity.

In addition, the paper lacks of some crucial aspects mainly from the chemical point of view.

Some suggestion are below reported.

Keywords: “modified Hyperbola model; nutrient release; nutrient releasing model”: this three keywords are not indipendent one each other, and they can be condensed in only two, e.g. nutrient release; modified Michaelis and Menten model.

Line 74: “This study…”, which one?

Lines 82-85: This sentence is too long and not clear. Furthermore, the novelty of this work must be better stressed.

Lines 89-90: “The biochar………..2017”: this sentence is not clear. Why it is necessary to report that analysis were performed in 2017?

Line 91: “The biochar made from pyrolysis system that was top to bottom method”: what is the meaning of this sentence?

Line 94: 1.5 m3 were loaded at the beginning of the test and the system was operated in batch mode?

Page 3, Table 1: TOC, TIC and TC of biochar do not perfectly match. Please check the reported values.

Line 98: the criteria adopted to select the ratios to be investigated are not reported.

Line 98 and Line 109: it seems that (6:4) ratio is mentioned but not tested

Lines 111-115: experimental conditions are not clear: this paragraph should be rewritten to allow a better understanding of the experimental procedure

Fig. 1 (ammonia accumulation in biochar): it is Accumulated or released? Please clarify. Furthermore, it seems that apart from (9:1) and (8:2), in all the other cases any steady condition was not achieved.

The same considerations for P, as in Fig. 2, and for K, as in Fig.3: in this last case a quick increase is observed after 20 days.

Author Response

Dear Dr. Reviewer 4

Thank you so  much for your valuable comments.

I modified my manuscript as you indicated.

All best,

JoungDu Shin 

Round  2

Reviewer 2 Report

The issues previously raised have been satisfactorily addressed.

Author Response

Dear Dr. Reviewer 2

Thank you so much for your comments.

I am going to submit the more upgraded version of manuscript.

All best,

JoungDu Shin

Reviewer 3 Report

Reviewer understands that authors do not have elemental composition at this time, however, in reviewer's perspective, the elemental composition (N, P, K and Si) of biochar is essential for meaningful understanding of how much percentage of the nutrients from biochar were released.  In fact, reviewer strongly recommend this experiment for pig manure as well. Reviewer do not recommend this article for publication without this data. The suggested elemental analysis (N, P, K and Si) experiment is sort of a control experiment for the scope of this project. Publishing scientific article without appropriate controls is not ethical.

Also reviewer sincerely apologize for the autocorrect errors: The word "national" in the original comments should be "rationale".  

Author Response

Dear Dr. Reviewer 3

Thank you so much for your comments.

I am going to submit the more upgraded version of manuscript.

I also modified Table 1 including elementar compositions as N, P, K, and Si

in biochar and pig manure compost.   

Reviewer 4 Report

Manuscript quality and overall significance have been substantially improved after the revision.

Authors successfully implemented all suggestions by the reviewer, thus reaching a quality worth of publication.

Author Response

Dear Dr. reviewer 4

Thank you so much for your comments.

I am going to submit the more upgraded version of manuscript.

All best,

JoungDu Shin

Round  3

Reviewer 3 Report

Reviewer agree to accept this article for publication in present form.